# DIFFERENCE BACK PROPAGATION WITH INVERSE SIGMOID FUNCTION

## ABSTRACT

Since the proposal of neural networks, the derivative-based back-propagation algorithm has been the default setting. However, the derivative for a nonlinear function is an approximation for the difference of the function values, and it would be a more precise way to do back propagation using the difference directly instead of the derivative. While the back propagation algorithm has been the rule-of-thumb for neural networks, it becomes one of the bottlenecks in modern large deep learning models. With the explosion of big data and large-scale deep learning models, a tiny change in back propagation could lead to a huge difference. Here we propose a new back propagation algorithm based on the inverse sigmoid function to calculate the difference instead of the derivative, and we verified its effectiveness with basic examples.

## 1 INTRODUCTION

Since the proposal of neural networks in 1943 (McCulloch & Pitts, 1943), the chain-rule back propagation (Dreyfus, 1962), based on derivatives, has been the only way to train neural network models. All parameters in neural networks were updated based on the derivatives of the cost function with respect to the parameters, calculated with a chain rule by traversing the network backward. To our knowledge, no new method for performing backpropagation has been proposed.

Recent years have witnessed great progress on big data as well as deep learning models. To name a few, there has been ImageNet dataset with 14 million labeled images released in 2015 (Russakovsky et al., 2015), Twitter100k with 100,000 image-text pairs released in 2017 (Hu et al., 2017), TextCaps with 145k captions for 28k images released in 2020 (Sidorov et al., 2020), and BuildingNet composed of 100k satellite images released in 2021 (Selvaraju et al., 2021). The size of datasets grows rapidly, and the size of deep learning models also explodes. The model has grown from relatively small machine learning models like the convolutional neural network (Fukushima & Miyake, 1982; LeCun et al., 1989) with thousands of parameters to BERT (Devlin et al., 2018) with 110 million parameters in 2018, and to V-MoE (Riquelme et al., 2021) with 15 billions parameters in 2021. Nevertheless, all these models have been using the same derivative-based back propagation algorithm. Although the models have shown great performance, it seems we are facing a bottleneck because nowadays we need to enlarge the models to billions of parameters to improve the accuracy by only a few percentages.

Trying to explore more options to break the bottleneck, here we propose a new back propagation method that makes a tiny change on the widely applied back propagation algorithm. The proposed difference back propagation calculates the gradient based on the difference instead of the derivative, making the gradients more reliable while propagating backward by maintaining the consistency of the activation function. In this paper, we experimented with the new back propagation method with sigmoid activation function in small neural networks to illustrate how it works. The algorithm is described in Sec. 2 and the performance is shown in Sec. 3.

## 2 METHOD

Our method only makes changes to the activation function. Here we assume all the other parts remain the same as a traditional neural network.

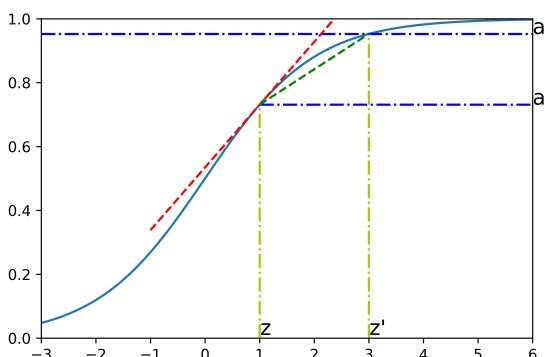

Figure 1: Illustration of the inconsistency of traditional back propagation. The change in $z$ is calculated based on the derivative (red dashed line), but the difference-based slope (green dashed line) would reflect the correspondence between $z$ and $a$ more precisely.

In the traditional way, the forward and backward propagations are calculated as Eq. 1 and Eq. 2:

$$a = sigmoid(z) = \frac{1}{1 + e^{-z}} \tag{1}$$

$$\frac{dl}{dz} = \frac{da}{dz}\frac{dl}{da} = a(1-a)\frac{dl}{da} \tag{2}$$

In which $z$ and $a$ are the neuron values before and after the activation function, respectively, and $l$ is the loss function. This chain rule works perfectly in the limit of learning rate approaching 0. However, with a finite learning rate, when $a$ is updated with Eq. 3, and the corresponding $z$ is updated with Eq. 4 which is not consistent with the changes on $a$.

$$a\_updated = a - \frac{dl}{da} * learning\_rate \tag{3}$$

$$z\_updated = z - \frac{dl}{dz} * learning\_rate \neq inv\_sig(a\_updated) \tag{4}$$

$$inv\_sig(a) = -log_e(1/a - 1) \tag{5}$$

This inconsistency is further illustrated in Fig. 1. When we perform backpropagation with optimization algorithms like gradient descent, the gradient of $a$ is first calculated based on the loss function, and then the gradient of $z$ is calculated with chain rule. When $a$ is updated to $a'$ based on the calculated gradients, $z$ is updated proportional to the slope of the red dashed line. However this line doesn't reflect the relationship between $a$ and $z$, instead, the green line indicates the corresponding $z'$ that satisfies the consistency: when $z$ changes to $z'$, the corresponding $a$ changes to $a'$.

We propose a new formula for the back propagation chain rule as in Eq. 6

$$\frac{dl}{dz} = \frac{\Delta a}{\Delta z}\frac{dl}{da} = \frac{a' - a}{z' - z}\frac{dl}{da} \tag{6}$$

In which $a' = a - learning\_rate * dl/da$, and $z' = inv_sig(a')$. We call this formula Difference Back Propagation (DBP) because it's calculated based on the difference of $z$ instead of the derivative. There are a few advantages of this method: 1) It's consistent and precise in terms of the changes of both $z$ and $a$, 2) It could avoid gradient vanishing from sigmoid function, and 3) DBP works not only for sigmoid activation function, but any function that has an inverse function, even for

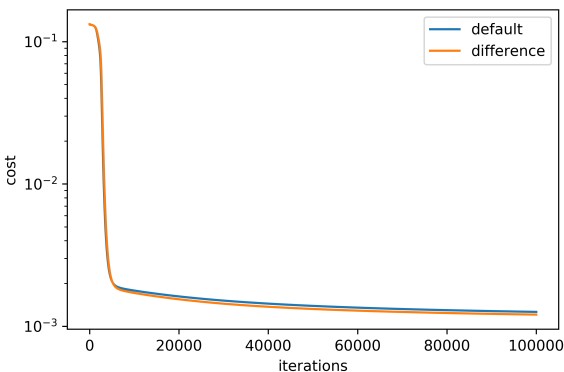

Figure 2: The cost functions with respect to training iterations for DBP and traditional back propagation in (1,2,1) neural network.

those functions that are not derivable or even continuous. For example, the derivative of leaky reLU activation function at 0 is not well defined, and with DBP we don't need to worry about it anymore.

Sigmoid activation function often faces a problem of vanishing gradient, because $a$ is too close to 1 and the derivative is too close to 0 when $z$ goes large. More specifically, with float64 precision, when $z > 36$, computers can not distinguish a from 1, resulting in a derivative of exact 0 during the calculation. With DBP, this issue is solved because we no longer calculate the derivative. However, there is still a minor issue with it because the definition domain of inverse sigmoid function is $(0, 1)$ exclusively, so we have to constrain $a$ strictly smaller than 1, resulting that the range of $z$ is no longer the entire set of real numbers. This problem can be solved by utilizing the Taylor Expansion and representing $a$ as $1 - x$ when $a$ is close to 1. This is beyond the scope of this paper, and we are setting a range constraint on $a$ along the experiments.

we conducted basic experiments to demonstrate the effectiveness of DBP, which are described in the next section. The code repository will be open-sourced later with respect to double-blind review.

## 3 RESULT

The DBP method has been experimented with small neural network models based on a small set of generative data.

The dataset consists of 100 random points with scaled cosine function. The inputs are numbers in range $(-1, 1)$, and outputs are numbers in range $(0, 1)$. The data is not split into train/test sets because the DBP method only affect the training process and the generalizability or over-fitting is not under consideration.

To check the effectiveness of DBP, we trained a neural network with only one hidden layer with 2 neurons. Including the input and output neurons, the neural network structure is $(1, 2, 1)$. The cost function is calculated as root mean squared error and the comparison between DBP and traditional back propagation is shown in Fig. 2. As DBP is a slight modification on the back propagation algorithm, the training costs are almost identical and the resulting performances are similar. However, there is a small but observable improvement for the convergence speed as well as the final cost, indicating that DBP is doing a better job than the traditional back propagation algorithm.

As mentioned before, a constraint is applied over the $a$ values during the training. $a$ is restricted to $(10^{-16}, 1 - 10^{-16})$. The upper bound is used to avoid overflow in the inverse sigmoid function and the lower bound is for the symmetry with the upper bound. In the meantime, the value of $z' - z$ is also restricted to avoid dividing by zero. Given that $z' - z$ is zero only when $a' - a$ is zero, so we force the zero values in $z' - z$ to 1 to make the slope zero.

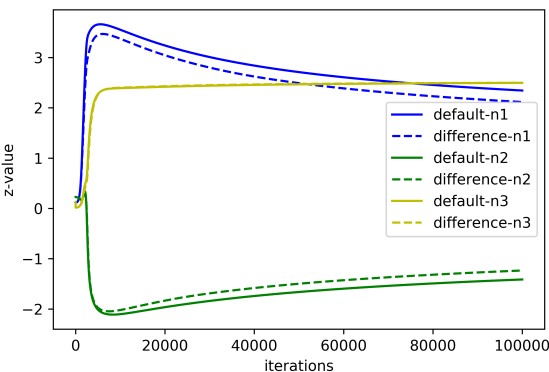

Figure 3: The neuron values $z$ with respect to the training iterations of three randomly selected samples, for DBP and traditional back propagation, in (1,2,1) neural network.

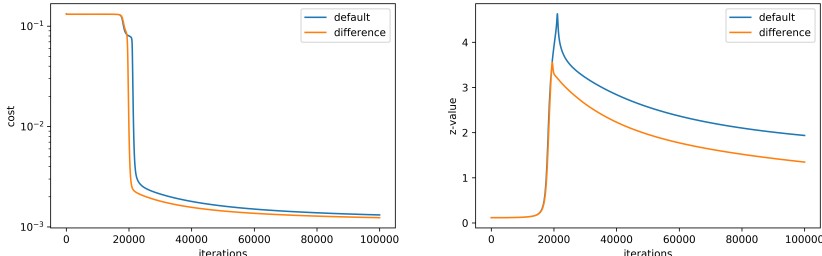

Figure 4: The cost functions and neuron values $z$ of a random sample with respect to the training iterations, for DBP and traditional back propagation, in (1,2,2,1) neural network.

To further investigate what the DBP algorithm does, the $z$ values of the neurons are visualized as in Fig.3. There are in total three neurons in the network (two hidden and one output). One data sample is randomly picked and the convergence of the neuron values are compared between the traditional back propagation and DBP. The two algorithms work almost the same way at the beginning of the model training, and the difference becomes significant only when the neuron values goes far away from zero. DBP prevents the neuron value $z$ from becoming too large or too small when the gradient disappears, because the gradient as in Eq. 6 is smaller than the traditional back propagation as in Eq. 2 when the updating direction is away from zero, and, on the contrary, larger in the case of toward zero.

Larger models are also experimented with. Fig. 4 shows the convergence of the cost function and neuron value $z$ of a random sample at a random neuron in the structure of the neural network $(1, 2, 2, 1)$. The results are very similar to the network $(1, 2, 1)$: with DBP, the cost function decays slightly faster and the neuron values tend not to go far from zero so that gradient vanishing is prevented.

Fig. 5 shows the convergence of loss function and accuracy of a transformer based classification model on news topic classification. with all the same hyperparameters (d_model=32, nlayers=2, nhead=4, ff=64), DBP showed clear advantage on both convergence speed and final performance in terms of accuracy (bottom two sub-figures are zoomed-in to show the difference).

## 4    CONCLUSION

We propose a new backpropagation algorithm, DBP, for neural networks, which is derived from inverse sigmoid function and the difference between the $z$ values. DBP has shown a better perfor-

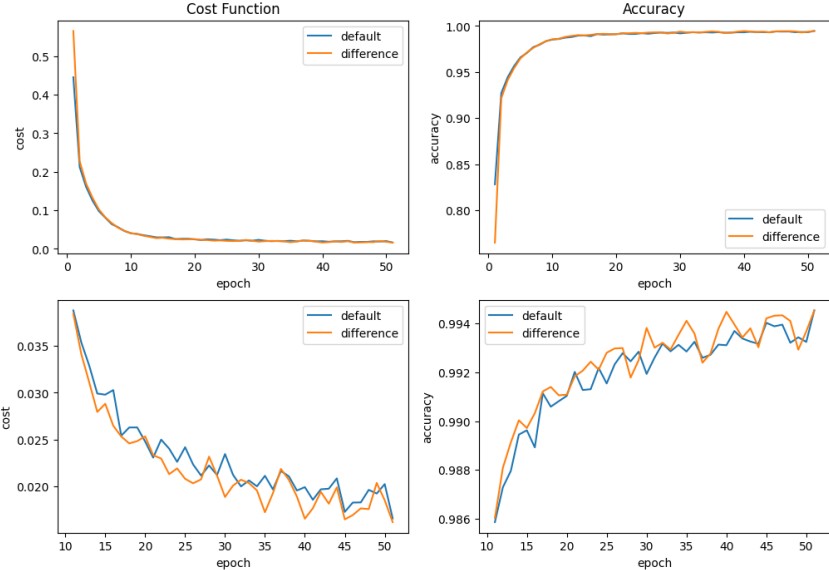

Figure 5: The loss function and model accuracy with respect to iterations, for DBP (difference) and traditional derivative back propagation (default), in basic transformer based classification using the AG News dataset from Hugging Face's datasets library with 4-category news topic classification

mance than the traditional derivative-based back propagation, as well as effectiveness in preventing gradient vanishing due to sigmoid function. We believe that DBP is a more accurate way to do back propagation because it maintains consistency between neuron values before and after the activation function.

The DBP algorithm can also be applied to other activation functions, as long as there is an inverse function accordingly. Without derivatives of the activation function, DBP allows for the applications of activation functions that are not derivable or continuous.

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
