# OpenReview forum: "Difference back propagation with inverse sigmoid function"
_ICLR.cc/2026/Conference — Submitted to ICLR 2026_

### Official Review · Reviewer_EQg7 · 2025-10-25

**Soundness:** 1
**Presentation:** 1
**Contribution:** 1
**Rating:** 0
**Confidence:** 5

**Summary:**

The paper proposes replacing the derivative of activation function with a difference quotient to fix an alleged inconsistency in backpropagation and claims this helps with vanishing gradients.

**Strengths:**

n/a

**Weaknesses:**

The described inconsistency in the paper does not exist. Gradient descent adjusts parameters, and activations a or pre-activations z are functions of parameters, so there is no inconsistency between updated z' and a'. Therefore, the paper is trying to solve a non-existent problem by replacing true gradient of activations by its finite difference approximation in backpropagation. Also, ignoring decades of prior work on training neural networks, makes the contribution appear uninformed.

The authors are strongly encouraged to further strengthen their conceptual understanding of deep learning and conduct a more thorough literature review before attempting to develop and publish new methods.

**Questions:**

n/a

---

### Official Review · Reviewer_Zs9A · 2025-11-01

**Soundness:** 1
**Presentation:** 1
**Contribution:** 1
**Rating:** 0
**Confidence:** 5

**Summary:**

The authors propose to modify the backpropagation equation in neural networks to improve training. They replace the derivative of the activation function (sigmoid here) by $(a-a')/(z-z')$. They observe improvements on neural networks with 3 and 5 neurons on a synthetic task.

**Strengths:**

The motivation is grounded as backpropagation is at the core of every model training.

**Weaknesses:**

- Line 29: "To our knowledge, no new method for performing backpropagation has been proposed.". The authors seem to be unaware of all the literature about alternatives to backpropagation. More generally, no related work is discussed, there are only 10 references and the most recent one is from 2021. I would advise the authors to look up "alternative to backpropagation arxiv" on any search engine.
- The maths are wrong. The authors assume that after a step the activation $a$ will move exactly in the direction of its gradient (Eq. 3), which is wrong. However it may be a good guess.
- I do not understand why the authors use the inverse sigmoid function to recompute $z$ from $a$ when it has already been computed during the forward pass.
- The experimental section is extremely lacking. A 3-neurons network on a synthetic task is not a good benchmark, we would expect at the very least larger networks (e.g. 2 hidden layers, dimension 128) on CIFAR10 for instance.

The paper is only 4.5 pages long, which leaves plenty of room to include more experiments, related works, etc.

**Questions:**

No question.

---

### Official Review · Reviewer_ZEzY · 2025-11-02

**Soundness:** 1
**Presentation:** 2
**Contribution:** 1
**Rating:** 0
**Confidence:** 4

**Summary:**

The submitted manuscript considers an alternative to the classical back-propagation (BP) for activations and proposes replacing it with a finite difference approximation. The obtained numerical results demonstrate that this approach leads to minor improvement in convergence for the considered toy models.

**Strengths:**

The submission is clearly written, and equations support the proposed approach.

**Weaknesses:**

I have identified many weaknesses in the submitted work and have listed the most crucial ones below.
1. The motivation for the proposed modification of the backpropagation (BP) is confusing. The exact gradient is essential for optimizers that update the model's parameters. If one approximates the gradient with a finite difference, then optimizers may converge to the wrong quasi-optimal parameters that do not correspond to the original problem. Moreover, typically only the stochastic gradient estimate is available in BP, and it remains unclear how this factor can be combined with the proposed approach.
2. No theoretical analysis or even intuition on how the proposed approach resolves the stated problem of "Although the models have shown great performance, it seems we are facing a bottleneck because nowadays we need to enlarge the models to billions of parameters to improve the accuracy by only a few percentages."
3. The proposed approach is empirically tested only for the toy models and toy datasets, which is insufficient to make any well-supported conclusion about its effectiveness.
4. Although generalization ability is crucial for deep learning models, I see that such analysis is explicitly excluded from the consideration.
5. The observed gain in cost (btw what is cost in the y-axis in Figures 2 and 4?)  looks not so large and can be explained with some random initialization. No proper statistical analysis of the significance of the presented gain is provided.

**Questions:**

1. Do you have any results for medium or large-scale models and datasets? E.g., ResNet18 and CIFAR10 and/or some standard benchmark in NLP like LLaMa or GPT-like models?
2. Backpropagation (BP) algorithms compute gradients of parameters to update them and minimize the loss function with a proper optimizer. Why have you considered the objective output of BP as an "inconsistency"? Inconsistency with respect to what?
3. What optimizers have you used to obtain the reported learning curves for BP and your approach?
4. Cross-entropy loss function incorporated sigmoid activation and avoided the mentioned instabilities since the sigmoid function is not computed standalone. So, which cases (specific application tasks) are suitable for your method?
5. How can you explain that there is no difference between your method and BP for the topic classification task presented in Figure 5? The zoomed plots do not convince of the stability of the gain since the noise is large. Averaging across multiple runs and plotting the standard deviation are necessary to support the conclusion.

---

### Meta-Review · Area_Chair_y6VN · 2026-01-06

**Summary:**

The paper suggests changes to the backpropagation algorithm by using difference of function values rather than derivatives. Overall, the paper received unanimously low scores (0, 0, 0), and no rebuttal was provided by the authors. All reviewers found multiple weaknesses in presentation, soundness and novelty. I encourage the authors to take the detailed feedback into account and submit a full-length improved manuscript in the future.

The paper is not recommended for acceptance in its current form.

**Reviewer Concerns:**

No rebuttal was provided.

**Reviewer Scores:**

Reviewers would have maintained their score (0).

---

### Decision · Program_Chairs · 2026-01-26

Reject